# An Efficient Membership Inference Attack for the Diffusion Model by Proximal Initialization

**Fei Kong**[1]   **Jinhao Duan**[2]   **Ruipeng Ma**[1]   **Hengtao Shen**[1]   **Xiaoshuang Shi**[1]
**Xiaofeng Zhu**[1]*   **Kaidi Xu**[2]*

[1]University of Electronic Science and Technology of China
[2]Drexel University

kong13661@outlook.com    kx46@drexel.edu

## Abstract

Recently, diffusion models have achieved remarkable success in generating tasks, including image and audio generation. However, like other generative models, diffusion models are prone to privacy issues. In this paper, we propose an efficient query-based membership inference attack (MIA), namely Proximal Initialization Attack (PIA), which utilizes groundtruth trajectory obtained by $\epsilon$ initialized in $t = 0$ and predicted point to infer memberships. Experimental results indicate that the proposed method can achieve competitive performance with only two queries that achieve at least $6\times$ efficiency than the previous SOTA baseline on both discrete-time and continuous-time diffusion models. Moreover, previous works on the privacy of diffusion models have focused on vision tasks without considering audio tasks. Therefore, we also explore the robustness of diffusion models to MIA in the text-to-speech (TTS) task, which is an audio generation task. To the best of our knowledge, this work is the first to study the robustness of diffusion models to MIA in the TTS task. Experimental results indicate that models with mel-spectrogram (image-like) output are vulnerable to MIA, while models with audio output are relatively robust to MIA. Code is available at `https://github.com/kong13661/PIA`.

## 1   Introduction

Recently, the diffusion model Ho et al. (2020); Song et al. (2021b); Song & Ermon (2019) has emerged as a powerful approach in the field of generative tasks, achieving notable success in image generation Rombach et al. (2022); Saharia et al. (2022), audio generation Popov et al. (2021); Kong et al. (2021), video generation Yang et al. (2022); Ho et al. (2022), and other domains. However, like other generative models such as GANs Goodfellow et al. (2020) and VAEs Kingma & Welling (2013); Xu et al. (2021), the diffusion model may also be exposed to privacy risks Bommasani et al. (2021) and copyright disputes Hristov (2016). Dangers such as privacy leaks Pham & Le (2020) and data reconstruction Zhang et al. (2020) may compromise the model. Recently, some researchers have explored this topic Duan et al. (2023); Matsumoto et al. (2023); Hu & Pang (2023); Carlini et al. (2023), demonstrating that diffusion models are also vulnerable to privacy issues.

Membership Inference Attacks (MIAs) are the most common privacy risks Shokri et al. (2017). MIAs can cause privacy concerns directly and can also contribute to privacy issues indirectly as part of data reconstruction. Given a pre-trained model, MIA aims to determine whether a sample is in the training set or not.

Generally speaking, MIA relies on the assumption that a model fits the training data better Yeom et al. (2018); Shokri et al. (2017), resulting in a smaller training loss. Recently, several MIA techniques have been proposed for diffusion models Duan et al. (2023); Matsumoto et al. (2023); Hu & Pang (2023). We refer to the query-based methods proposed in Matsumoto et al. (2023); Hu & Pang (2023) as Naive Attacks because they directly employ the training loss for the attack. However, unlike GANs or VAEs, the training loss for diffusion models is not deterministic because it requires the generation of Gaussian noise. The random Gaussian noise may not be the one in which diffusion models fit best.

---

*Equal corresponding author

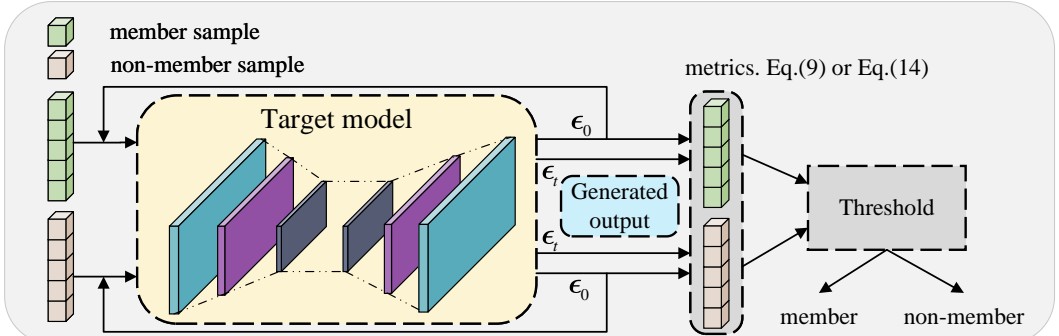

Figure 1: A overview of PIA. First, a sample is an input into the target model to generate $\epsilon$ at time $0$. Next, we combine the original sample with $\epsilon_0$ and input them into the target model to generate $\epsilon$ at time $t$. After that, we input all three variables into a metric and use a threshold to determine if the sample belongs to the training set.

This can negatively impact the performance of the MIA attack. To address this issue, the concurrent work SecMI Duan et al. (2023) adopts an iterative approach to obtain the deterministic $x$ at a specific time $t$, but this requires more queries, resulting in longer attack times. As models grow larger, the time required for the attack also increases, making time an important metric to consider.

To reduce the time consumption, inspired by DDIM and SecMI, we proposed a Proximal Initialization Attack (PIA) method, which derives its name from the fact that we utilize the diffusion model's output at time $t = 0$ as the noise $\epsilon$. PIA is a query-based MIA that relies solely on the inference results and can be applied not only to discrete time diffusion models Ho et al. (2020); Rombach et al. (2022) but also to continuous time diffusion models Song et al. (2021b). We evaluate the effectiveness of our method on three image datasets, CIFAR10 Krizhevsky et al. (2009), CIFAR100 and TinyImageNet for DDPM and on two images dataset, COCO2017 Lin et al. (2014) and Laion5B Schuhmann et al. (2022) for Stable DIffuion, as well as three audio datasets, LJSpeech Ito & Johnson (2017), VCTK Junichi et al. (2019), and LibriTTS Zen et al. (2019).

To our knowledge, recent research on MIA of diffusion models has only focused on image data, and there has been no exploration of diffusion models in the audio domain. However, audio, such as music, encounters similar copyright and privacy concerns as those in the image domain CNN (2022); WashingtonPost (2022). Therefore, it is essential to conduct privacy research in the audio domain to determine whether audio data is also vulnerable to attacks and to identify which types of diffusion models are more robust against privacy attacks. To investigate the robustness of MIA on audio data, we conduct experiments using Naive Attack, SecMI Duan et al. (2023), and our proposed method on three audio models: Grad-TTS Popov et al. (2021), DiffWave Kong et al. (2021), and FastDiff Huang et al. (2022). The results suggest that the robustness of MIA on audio depends on the output type of the model.

Our contributions can be summarized as follows:

- We propose a query-based MIA method called PIA. Our method employs the output at $t = 0$ as the initial noise and the errors between the forward and backward processes as the attack metric. We generalize the PIA on both discrete-time and continuous-time diffusion models.

- Our study is the first to evaluate the robustness of MIA on audio data. We evaluate the robustness of MIA on three TTS models (Grad-TTS, DiffWave, FastDiff) and three TTS datasets (LJSpeech, VCTK, Libritts) using Naive Attack, SecMI, and our proposed method.

- Our evaluations show that PIA matches SecMI's AUC performance and outperforms it in TPR @ 1% FPR, while being 5-10 times faster. Moreover, our data imply that in text-to-speech tasks, models producing audio are more resistant to MIA attacks than those generating image-like mel-spectrograms. We therefore suggest using audio-output generation models to minimize privacy risks in audio creation tasks.

## 2 RELATED WORKS AND BACKGROUND

**Generative Diffusion Models**  Generative diffusion models have recently achieved significant success in both image Ramesh et al. (2022); Rombach et al. (2022); Yuan et al. (2023a) and audio generation tasks Huang et al. (2022); Chen et al. (2021); Popov et al. (2021). Unlike GANs Goodfellow et al. (2020); Yuan & Moghaddam (2020); Yuan et al. (2023b), which consist of a generator and a discriminator, diffusion models generate samples by fitting the inverse process of a diffusion process from Gaussian noise. Compared to GANs, diffusion models typically produce higher quality samples and avoid issues such as checkerboard artifacts Salimans et al. (2016); Donahue et al. (2017); Dumoulin et al. (2017). A diffusion process is defined as $\boldsymbol{x}_t = \sqrt{\alpha_t}\boldsymbol{x}_{t-1} + \sqrt{\beta_t}\boldsymbol{\epsilon}_t, \boldsymbol{\epsilon}_t \sim \mathcal{N}(\mathbf{0}, \mathbf{I})$, where $\alpha_t + \beta_t = 1$ and $\beta_t$ increases gradually as $t$ increases, so that eventually, $\boldsymbol{x}_t$ approximates a random Gaussian noise. In the reverse diffusion process, $\boldsymbol{x}_t'$ still follows a Gaussian distribution, assuming the variance remains the same as in the forward diffusion process, and the mean is defined as $\tilde{\boldsymbol{\mu}}_t = \frac{1}{\sqrt{a_t}}\left(\boldsymbol{x}_t - \frac{\beta_t}{\sqrt{1-\bar{a}_t}}\bar{\boldsymbol{\epsilon}}_\theta(\boldsymbol{x}_t, t)\right)$, where $\bar{\alpha}_t = \prod_{k=0}^{t}\alpha_k$ and $\bar{\alpha}_t + \bar{\beta}_t = 1$. The reverse diffusion process becomes $\boldsymbol{x}_{t-1} = \tilde{\boldsymbol{\mu}}_t + \sqrt{\beta_t}\boldsymbol{\epsilon}, \boldsymbol{\epsilon} \sim \mathcal{N}(\mathbf{0}, \mathbf{I})$. One can obtain a loss function Eq. (1) by minimizing the distance between the predicted and groundtruth distributions. Song et al. (2021b) transforms the discrete-time diffusion process into a continuous-time process and uses SDE ( Stochastic Differential Equation) to express the diffusion process. To accelerate the generation process, several methods have been proposed, such as Salimans & Ho (2022); Dockhorn et al. (2022); Xiao et al. (2022). DDIM Song et al. (2021a) is another popular method that proposes a forward process different from diffusion process with the same loss function as DDPM, allowing it to reuse the model trained by DDPM while achieving higher generation speed.

$$L = \mathbb{E}_{x_0, \bar{\epsilon}_t}\left[\left\|\bar{\boldsymbol{\epsilon}}_t - \boldsymbol{\epsilon}_\theta\left(\sqrt{\bar{\alpha}_t}x_0 + \sqrt{1-\bar{\alpha}_t}\bar{\boldsymbol{\epsilon}}_t, t\right)\right\|^2\right]. \tag{1}$$

**Membership Inference Privacy**  Different from conventional adversarial attacks Xu et al. (2018; 2020); Zhang et al. (2022), Membership inference attack (MIA) Shokri et al. (2017) aims to determine whether a sample is part of the training data. It can be formally described as follows: given two sets, the training set $\mathcal{D}_t$ and the hold-out set $\mathcal{D}_h$, a target model $m$, and a sample $\boldsymbol{x}$ that either belongs to $\mathcal{D}_t$ or $\mathcal{D}_h$, the goal of MIA is to find a classifier or function $f(\boldsymbol{x}, m)$ that determines which set $\boldsymbol{x}$ belongs to, with $f(\boldsymbol{x}, m) \in \{0, 1\}$ and $f(\boldsymbol{x}, m) = 1$ indicating that $\boldsymbol{x} \in \mathcal{D}_t$ and $f(\boldsymbol{x}, m) = 0$ indicating that $\boldsymbol{x} \in \mathcal{D}_h$. If a membership inference attack method utilizes a model's output obtained through queries to attack the model, it is called query-based attackDuan et al. (2023); Matsumoto et al. (2023); Hu & Pang (2023). Typically, MIA is based on the assumption that training data has a smaller loss compared to hold-out data. MIA for generation tasks, such as GANs Pham & Le (2020) and VAEs Hilprecht et al. (2019); Chen et al. (2020), has also been extensively researched.

Recently, several MIA methods designed for diffusion models have been proposed. Matsumoto et al. (2023) proposed a method that directly employs the training loss Eq. (1) and find a specific $t$ with maximum distinguishability. Because they directly use the training loss, we refer to this method as Naive Attack. SecMI Duan et al. (2023) improves the attack effectiveness by iteratively computing the $t$-error, which is the error between the DDIM sampling process and the inverse sampling process at a certain moment $t$.

**Threat model**  We follow the same threat model as Duan et al. (2023), which needs to access intermediate outputs of diffusion models. This is a query-based attack without the knowledge of model parameters but not fully end-to-end black-box. In scenarios such as inpainting Lugmayr et al. (2022), and classification Li et al. (2023), they also employ the intermediate output of the diffusion model. These works utilize a pre-trained model on a huge dataset to do other tasks, such as inpainting, and classification without fine-tuning. To meet these requirements, future service providers might consider opening up APIs for intermediate outputs. Our work is applicable to such scenarios.

## 3 METHODOLOGY

In this section, we introduce DDIM, a variant of DDPM, and provide a proof that if we know any two points in the DDIM framework, $\boldsymbol{x}_k$ and $\boldsymbol{x}_0$, we can determine any other point $\boldsymbol{x}_t$. We then propose a new MIA method that utilizes this property to efficiently obtain $\boldsymbol{x}_{t-t'}$ and its corresponding predicted sample $x'_{t-t'}$. We compute the difference between these two points and use it to determine if a sample

is in the training set. Specifically, samples with small differences are more likely to belong to the training set. An overview of this proposed method is shown in Fig. 1.

## 3.1 PRELIMINARY

**Denoising Diffusion Implicit Models** To accelerate the inference process of diffusion models, DDIM defines a new process that shares the same loss function as DDPM. Unlike the DDPM process, which adds noise from $x_0$ to $x_T$, DDIM defines a diffusion process from $x_T$ to $x_1$ by using $x_0$. The process is described in Eq. (2) and Eq. (3). The distribution $q_\sigma(\boldsymbol{x}_T \mid \boldsymbol{x}_0)$ is the same as in DDPM.

$$q_\sigma(\boldsymbol{x}_{1:T} \mid \boldsymbol{x}_0) := q_\sigma(\boldsymbol{x}_T \mid \boldsymbol{x}_0) \prod_{t=2}^{T} q_\sigma(\boldsymbol{x}_{t-1} \mid \boldsymbol{x}_t, \boldsymbol{x}_0),  \tag{2}$$

$$q_\sigma(\boldsymbol{x}_{t-1} \mid \boldsymbol{x}_t, \boldsymbol{x}_0) = \mathcal{N}\left(\sqrt{\bar{\alpha}_{t-1}}x_0 + \sqrt{1 - \bar{\alpha}_{t-1} - \sigma_t^2} \cdot \frac{\boldsymbol{x}_t - \sqrt{\bar{\alpha}_t}x_0}{\sqrt{1 - \bar{\alpha}_t}}, \sigma_t^2\boldsymbol{I}\right). \tag{3}$$

The denoising process defined by DDIM is described below:

$$\begin{aligned} p(\boldsymbol{x}_{t'} \mid \boldsymbol{x}_t) &= p(\boldsymbol{x}_{t'} \mid \boldsymbol{x}_t, \boldsymbol{x}_0 = \overline{\boldsymbol{\mu}}(\boldsymbol{x}_t)) \\ &= \mathcal{N}\left(\boldsymbol{x}_{t'}; \frac{\sqrt{\bar{\alpha}_{t'}}}{\sqrt{\bar{\alpha}_t}}\left(\boldsymbol{x}_t - \left(\sqrt{1 - \bar{\alpha}_t} - \frac{\sqrt{\bar{\alpha}_t}}{\sqrt{\bar{\alpha}_{t'}}}\sqrt{1 - \bar{\alpha}_{t'} - \sigma_t^2}\right)\boldsymbol{\epsilon_\theta}(\boldsymbol{x}_t, t)\right), \sigma_t^2\boldsymbol{I}\right) \end{aligned} \tag{4}$$

## 3.2 FINDING GROUNDTRUTH TRAJECTORY

In this section, we will first demonstrate that if we know $\boldsymbol{x}_k$ and $\boldsymbol{x}_0$, we can determine any other $\boldsymbol{x}_t$. Then, we will provide the method for obtaining $\boldsymbol{x}_k$.

**Theorem 1** *The trajectory of $\{\boldsymbol{x}_t\}$ is determined if we know $x_0$ and any other point $x_k$ when $\sigma_t = 0$ under DDIM framework.*

**Proof** *In DDIM definition, if standard deviation $\sigma_t = 0$, the process adding noise becomes determined. So Eq. (3) can be rewritten to Eq. (5).*

$$\boldsymbol{x}_{t-1} = \sqrt{\bar{\alpha}_{t-1}}\boldsymbol{x}_0 + \sqrt{1 - \bar{\alpha}_{t-1}} \cdot \frac{\boldsymbol{x}_t - \sqrt{\bar{\alpha}_t}\boldsymbol{x}_0}{\sqrt{1 - \bar{\alpha}_t}}. \tag{5}$$

*Assuming that we know any point $\boldsymbol{x}_k$. Eq. (5) can be rewritten as $\frac{\boldsymbol{x}_{t-1} - \sqrt{\bar{\alpha}_{t-1}}\boldsymbol{x}_0}{\sqrt{1 - \bar{\alpha}_{t-1}}} = \frac{\boldsymbol{x}_t - \sqrt{\bar{\alpha}_t}\boldsymbol{x}_0}{\sqrt{1 - \bar{\alpha}_t}}$. By applying this equation recurrently, we can obtain Eq. (6). In other words, we can obtain any point $x_t$ except $x_k$.*

$$\boldsymbol{x}_t = \sqrt{\bar{\alpha}_t}\boldsymbol{x}_0 + \sqrt{1 - \bar{\alpha}_t} \cdot \frac{\boldsymbol{x}_k - \sqrt{\bar{\alpha}_k}\boldsymbol{x}_0}{\sqrt{1 - \bar{\alpha}_k}}. \tag{6}$$

We call the trajectory obtained from $\boldsymbol{x}_k$ *groundtruth trajectory*.

Assuming that the point is $\boldsymbol{x}_k = \sqrt{\bar{a}_k}\boldsymbol{x}_0 + \sqrt{1 - \bar{a}_k}\bar{\boldsymbol{\epsilon}}_k$, to find a better groundtruth trajectory, we choose $k = 0$ since the choice of $k$ is arbitrary, and approximate $\bar{\epsilon}_0$ using Eq. (7).

$$\boldsymbol{\epsilon_\theta}\left(\sqrt{\bar{a}_0}\boldsymbol{x}_0 + \sqrt{1 - \bar{a}_0}\bar{\boldsymbol{\epsilon}}_0, 0\right) \approx \boldsymbol{\epsilon_\theta}(\boldsymbol{x}_0, 0). \tag{7}$$

This choice is intuitive. First, $\bar{\alpha}_0$ is very close to 1, making the approximation in Eq. (7) valid. Second, the time $t = 0$ is the closest timing to the original sample, so the model is likely to fit it better.

## 3.3 EXPOSING MEMBERSHIP VIA GROUNDTRUTH TRAJECTORY AND PREDICTED POINT

Our approach assumes that the training set's samples have a smaller loss, similar to many other MIAs, meaning that the training samples align more closely with the groundtruth trajectory. We measure the distance between any groundtruth point $\boldsymbol{x}_{t-t'}$ and the predicted point $\boldsymbol{x}'_{t-t'}$ using the $\ell_p$-norm, which can be expressed by Eq. (8). Here, $\boldsymbol{x}'_{t-t'}$ denotes the point predicted by the model from $\boldsymbol{x}_t$. To apply this attack, we need to select a specific time $t - t'$, and we choose the time $t' = t - 1$ since it is

the closest. However, we will demonstrate later that the choice of $t'$ is not significant in discrete-time diffusion.

$$d_{t-t'} = \left\| \boldsymbol{x}_{t-t'} - \boldsymbol{x}'_{t-t'} \right\|_p. \tag{8}$$

To predict $\boldsymbol{x}'_{t-t'}$ from the groundtruth point $\boldsymbol{x}_t$, we apply the deterministic version ($\sigma_t = 0$) of the DDIM denoising process Eq. (4).

We use method described in Section 3.2 to obtain the groundtruth point $\boldsymbol{x}_t$ and $\boldsymbol{x}_{t-t'}$. We then insert these points into Eq. (8), giving us a simpler formula:

$$\frac{\sqrt{1 - \bar{\alpha}_{t-t'}}\sqrt{\bar{\alpha}_t} - \sqrt{1 - \bar{\alpha}_t}\sqrt{\bar{\alpha}_{t-t'}}}{\sqrt{\bar{\alpha}_t}} \left\| \bar{\boldsymbol{\epsilon}}_0 - \boldsymbol{\epsilon}_{\boldsymbol{\theta}} \left( \sqrt{\bar{a}_t}\boldsymbol{x}_0 + \sqrt{1 - \bar{a}_t}\bar{\boldsymbol{\epsilon}}_0, t \right) \right\|_p.$$

If we ignore the coefficient, $t'$ disappears. Finally, the metric ignoring the coefficient reduces to Eq. (9), where samples with smaller $R_{t,p}$ are more likely to be training samples.

$$R_{t,p} = \left\| \boldsymbol{\epsilon}_{\boldsymbol{\theta}} \left( \boldsymbol{x}_0, 0 \right) - \boldsymbol{\epsilon}_{\boldsymbol{\theta}} \left( \sqrt{\bar{a}_t}\boldsymbol{x}_0 + \sqrt{1 - \bar{a}_t}\boldsymbol{\epsilon}_{\boldsymbol{\theta}} \left( \boldsymbol{x}_0, 0 \right), t \right) \right\|_p. \tag{9}$$

Since $\epsilon$ is initialized in time $t = 0$, we call our method Proximal Initialization Attack (PIA).

**Normalization** The values of $\boldsymbol{\epsilon}_{\boldsymbol{\theta}} \left( \boldsymbol{x}_0, 0 \right)$ may not conform to a standard normal distribution, so we use Eq. (10) to normalize them. $N$ represents the number of elements in the sample, such as $h \times w$ for an image. We refer to this method as PIAN (PIA Normalized). Although this normalization cannot guarantee that $\hat{\boldsymbol{\epsilon}}_{\boldsymbol{\theta}} \left( \boldsymbol{x}_0, 0 \right) \sim \mathcal{N}(\boldsymbol{0}, \mathbf{I})$, we deem it reasonable since each element of $\bar{\epsilon}_t$ in the training loss Eq. (1) is identically and independently distributed.

$$\hat{\boldsymbol{\epsilon}}_{\boldsymbol{\theta}} \left( \boldsymbol{x}_0, 0 \right) = \frac{\boldsymbol{\epsilon}_{\boldsymbol{\theta}}(\boldsymbol{x}_0, 0)}{\mathbb{E}_{x \sim \mathcal{N}(0,1)}(|x|)\frac{\|\boldsymbol{\epsilon}_{\boldsymbol{\theta}}(\boldsymbol{x}_0, 0)\|_1}{N}} = N\sqrt{\frac{\pi}{2}} \frac{\boldsymbol{\epsilon}_{\boldsymbol{\theta}}(\boldsymbol{x}_0, 0)}{\|\boldsymbol{\epsilon}_{\boldsymbol{\theta}}(\boldsymbol{x}_0, 0)\|_1}. \tag{10}$$

To apply our attack, we first evaluate the value of $R_{t,p}$ on a sample, and use an indicator function:

$$f(\boldsymbol{x}, m) = \mathbb{1}[R_{t,p} < \tau]. \tag{11}$$

This indicator means we consider whether a sample is in the training set if $R_{t,p}$ is smaller than a threshold $\tau$. $R_{t,p}$ is obtained from $\boldsymbol{\epsilon}_{\boldsymbol{\theta}} \left( \boldsymbol{x}_0, 0 \right)$ (PIA) or $\hat{\boldsymbol{\epsilon}}_{\boldsymbol{\theta}} \left( \boldsymbol{x}_0, 0 \right)$ (PIAN).

### 3.4 FOR CONTINUOUS-TIME DIFFUSION MODEL

Recently, some diffusion models are trained with continuous time. As demonstrated in Song et al. (2021b), the diffusion process with continuous time can be defined by a stochastic differential equation (SDE) as $d\boldsymbol{x}_t = \boldsymbol{f}_t(\boldsymbol{x}_t)dt + g_t d\boldsymbol{w}_t$, where $\boldsymbol{w}_t$ is a Brownian process. One of the reverse processes is $d\boldsymbol{x}_t = \left( \boldsymbol{f}_t(\boldsymbol{x}_t) - \frac{1}{2} \left( g_t^2 + \sigma_t^2 \right) \nabla_{\boldsymbol{x}_t} \log p_t(\boldsymbol{x}_t) \right) dt + \sigma_t d\boldsymbol{w}$. When $\sigma_t = 0$, this formula becomes an ordinary differential equation (ODE): $d\boldsymbol{x}_t = \left( \boldsymbol{f}_t(\boldsymbol{x}_t) - \frac{1}{2} g_t^2 \nabla_{\boldsymbol{x}_t} \log p_t(\boldsymbol{x}_t) \right) dt$. Continuous-time diffusion model train an $\boldsymbol{s}_{\boldsymbol{\theta}}$ to approximate $\nabla_{\boldsymbol{x}_t} \log p_t(\boldsymbol{x}_t)$, so the loss function will be:

$$L = \mathbb{E}_{\boldsymbol{x}_0, \boldsymbol{x}_t \sim p(\boldsymbol{x}_t|\boldsymbol{x}_0)\bar{p}(\boldsymbol{x}_0)} \left[ \left\| \boldsymbol{s}_{\boldsymbol{\theta}} \left( \boldsymbol{x}_t, t \right) - \nabla_{\boldsymbol{x}_t} \log p \left( \boldsymbol{x}_t \mid \boldsymbol{x}_0 \right) \right\|^2 \right].$$

Replacing $\nabla_{\boldsymbol{x}_t} \log p_t(\boldsymbol{x}_t)$ with $\boldsymbol{s}_{\boldsymbol{\theta}} \left( \boldsymbol{x}_t, t \right)$, the inference procedure become the following equation:

$$d\boldsymbol{x}_t = \left( \boldsymbol{f}_t(\boldsymbol{x}_t) - \frac{1}{2} g_t^2 \boldsymbol{s}_{\boldsymbol{\theta}}(\boldsymbol{x}_t, t) \right) dt. \tag{12}$$

The distribution $p(\boldsymbol{x}_t|\boldsymbol{x}_0)$ is typically set to be the same as in DDPM for continuous-time diffusion models. Therefore, the loss of the continuous-time diffusion model and the loss of the concrete-diffusion model Eq. (1) are similar. Since DDPM and the diffusion model described by SDE share a similar loss, our method can be applied to continuous-time diffusion models. However, due to the different diffusion process, $R_{t,p}$ differs from Eq. (9). From Eq. (12), we obtain the following equation: $\boldsymbol{x}_{t-t'} - \boldsymbol{x}_t \approx d\boldsymbol{x}_t = \left( \boldsymbol{f}_t(\boldsymbol{x}_t) - \frac{1}{2} g_t^2 \boldsymbol{s}_{\boldsymbol{\theta}} \left( \boldsymbol{x}_t, t \right) \right) dt$. By substituting this equation into Eq. (8), we obtain the following equation:

$$\left\| \boldsymbol{x}_{t-t'} - \boldsymbol{x}'_{t-t'} \right\|_p \approx \left\| \left( \boldsymbol{f}_t(\boldsymbol{x}_t) - \frac{1}{2} g_t^2 \boldsymbol{s}_{\boldsymbol{\theta}} \left( \boldsymbol{x}_t, t \right) \right) dt + \boldsymbol{x}_t - \boldsymbol{x}'_{t-t'} \right\|_p. \tag{13}$$

Table 1: Performance of different methods on Grad-TTS. TPR@x% is the abbreviation for TPR@x% FPR.

| Method | LJspeech | | VCTK | | LibriTTS | | Query |
| --- | --- | --- | --- | --- | --- | --- | --- |
| | AUC | TPR@1% FPR | AUC | TPR@1% FPR | AUC | TPR@1% FPR | |
| NA Matsumoto et al. (2023) | 99.4 | 93.6 | 83.4 | 6.1 | 90.2 | 9.1 | **1** |
| SecMI Duan et al. (2023) | 99.5 | 94.0 | 87.0 | 14.8 | 93.9 | 19.7 | 60+2 |
| PIA | **99.6** | 94.2 | 87.8 | **20.6** | **95.4** | 30.0 | 1+1 |
| PIAN | 99.3 | **95.7** | **88.1** | 19.6 | 93.4 | **44.7** | 1+1 |

Table 2: Performance of the different methods on DDPM.

| Method | CIFAR10 | | TN-IN | | CIFAR100 | | Query |
| --- | --- | --- | --- | --- | --- | --- | --- |
| | AUC | TPR@1% FPR | AUC | TPR@1% FPR | AUC | TPR@1% FPR | |
| NA | 84.7 | 6.85 | 84.9 | 10.0 | 82.3 | 9.6 | **1** |
| SecMI | 88.1 | 9.11 | 89.4 | 12.7 | 87.6 | 11.1 | 10+2 |
| PIA | **88.5** | 13.7 | **89.6** | 17.1 | **89.4** | 19.6 | 1+1 |
| PIAN | 87.8 | **31.2** | 88.2 | **32.8** | 86.5 | **22.2** | 1+1 |

Solving this ODE incurs a truncation error that is positively correlated with $\Delta t$. Therefore, we take the limit as $t' \to 0$. In this case, higher-order infinitesimals can be neglected in Eq. (13). so, we can obtain $\|x_{t-t'} - x'_{t-t'}\|_p \approx \left\| \left( f_t(x_t) - \frac{1}{2} g_t^2 s_\theta (x_t, t) \right) \right\|_p dt$. Since the $t'$ is same when comparing two sampling, we can neglect $dt$ and use the following attack metric:

$$R_{t,p} = \left\| f_t(x_t) - \frac{1}{2} g_t^2 s_\theta (x_t, t) \right\|_p, \tag{14}$$

where $x_t$ is obtained from the output of $s_\theta(x_0, 0)$, similar to the discrete-time diffusion case.

## 4 EXPERIMENT

In this section, we evaluate the performance of PIA and PIAN and robustness of TTS models across various datasets and settings. The detailed experimental settings, including datasets, models, and hyper-parameter settings can be found in Appendix B.

### 4.1 EVALUATION METRICS

We follow the most convincing metrics used in MIAs Carlini et al. (2023), including AUC, the True Positive Rate (TPR) when the False Positive Rate (FPR) is 1%, i.e., TPR @ 1% FPR, and TPR @ 0.1% FPR.

### 4.2 PROXIMAL INITIALIZATION ATTACK PERFORMANCE

We train TTS models on the LJSpeech, VCTK, and LibriTTS datasets. We summarize the AUC and TPR @ 1% FPR results on GradTTS, a continuous-time diffusion model, in Table 1. We employ NA to denote Naive Attack. Compared to SecMI, PIA and PIAN achieve slightly better AUC performance, and significantly higher TPR @ 1% FPR performance, i.e., 5.4% higher for PIA and 10.5% higher for PIAN on average. However, our proposed method only requires $1 + 1$ queries, just one more query

Table 3: Performance of different methods on stable diffusion.

| Method | Laion5 | | Laion5 w/o text | | Laion5 Blip text | | Query |
| --- | --- | --- | --- | --- | --- | --- | --- |
| | AUC | TPR@1% FPR | AUC | TPR@1% FPR | AUC | TPR@1% FPR | |
| NA | 66.3 | 14.8 | 65.2 | 13.3 | 68.2 | 16.2 | **1** |
| SecMI | 69.1 | 16.1 | 71.6 | 14.5 | 71.6 | 17.8 | 10+2 |
| PIA | **70.5** | **18.1** | **73.9** | 19.8 | **73.3** | **20.2** | 1+1 |
| PIAN | 56.7 | 4.8 | 58.8 | 3.2 | 55.3 | 3.2 | 1+1 |

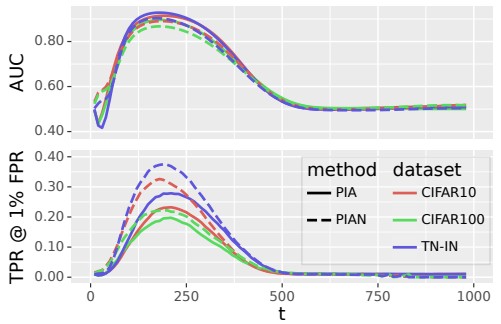 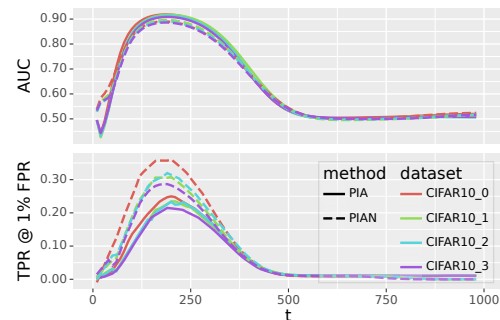

(a) The results of PIA and PIAN on DDPM for different values of t and different datasets.

(b) The results of PIA and PIAN on DDPM for different values of $t$ and different CIFAR10 splits.

Figure 2: The performance of PIA and PIAN as $t$ varies. The top row shows the results for AUC, and the bottom row shows the results for TPR @ 1% FPR.

than Naive Attack, and has a computational consumption of only 3.2% of SecMI. Both methods outperform SecMI and Naive Attack.

For DDPM, a discrete-time diffusion model, we present the results in Table 2. For this model, PIA performs slightly better than SecMI in terms of AUC but has a distinctly higher TPR @ 1% FPR than SecMI, i.e. 5.8% higher on average than SecMI. For PIAN, the AUC performance is slightly lower than PIA, but higher than SecMI, and the TPR @ 1% FPR performance is significantly better than SecMI, i.e. 17.8% higher on average than SecMI. Similar to the previous case, our attack only requires two queries on DDPM and the computational consumption is 17% of SecMI. Both methods outperform SecMI and Naive Attack.

For stable diffusion, we present the results in Table 3. We evaluated stable diffusion on Laion5 (training dataset) and COCO (evaluation dataset). Details are put into A.2. We tested three scenarios: knowing the ground truth text (Laion5), not knowing the ground truth text (Laion5 w/o text), and generating text through blip (Laion5 Blip text). PIA achieved the best results. PIA performs slightly better than SecMI in terms of AUC, i.e. 1.8% higher on average, but has a distinctly higher TPR @ 1% FPR than SecMI, i.e. 3.2% higher on average. Besides, our attack only requires two queries on DDPM and the computational consumption is 17% of SecMI.

However, PIAN does not work well in stable diffusion. PIAN based on the fact that we added noise that follows a normal distribution during training, and we use Eq. (10) to rescale the $\epsilon$ to normal distribution. However, rescaling is a rough operation and may not always transform into a normal distribution. Thus, some other transforms might have better performance. Additionally, the model's output might be more accurate before the rescaling.

We highly recommend using PIA as the preferred method for conducting attacks, because it is directly derived. It will always yield the desired results. But PIAN can be another choice, since it has better performance at TPR @ 1% FPR metric than PIA on some models.

### 4.3 ABLATION STUDY

Our proposed method has three hyper-parameters: $t$ and the $\ell_p$-norm used in the attack metrics $R_{t,p}$ presented in Eqs. (9) and (14). The threshold $\tau$ presented in Eq. (11).

**Impact of** $t$  To evaluate the impact of $t$, we attack the target model at intervals of $0.01 \times T$ from 0 to $T$ and report the results across different models and datasets.

We demonstrate the performance of our proposed method on two different models: GradTTS, a continuous-time diffusion model used for audio; and DDPM, a discrete-time diffusion model employed for images. The results indicate that our method produces a consistent pattern in the same model across different datasets, whether PIA or PIAN. Specifically, for DDPM, both AUC and TPR @ 1% FPR exhibit a rapid increase at the beginning as $t$ increases followed by a decline around $t = 200$ from Fig. 2a. In Fig. 2b, we randomly partition the CIFAR10 dataset four times and compare the performance of each partition. Consistent with the previous results, our method exhibits a similar

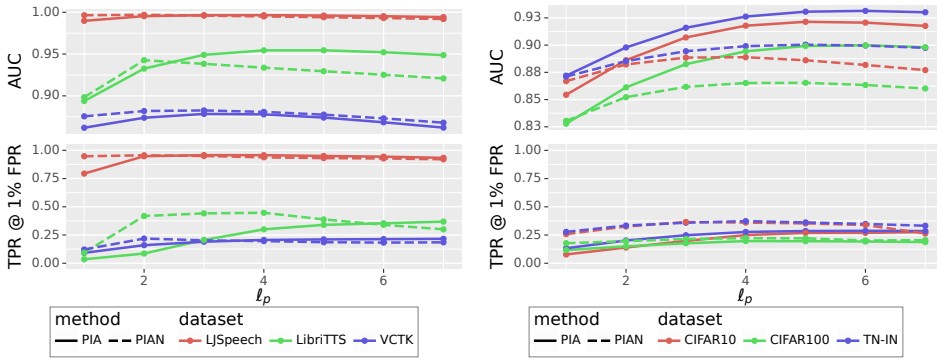

(a) The results of PIA and PIAN on Grad-TTS for different values of $\ell_p$-norm.

(b) The results of PIA and PIAN on DDPM for different values of $\ell_p$-norm.

Figure 3: The performance of our method as $\ell_p$-norm varies. The top row shows the results for AUC, and the bottom row displays the results for TPR @ 1% FPR.

Table 4: The variation of Attack Success Rate (ASR) and TPR/FPR on the victim model with the threshold determined by the surrogate model.

| | PIA | | | | PIAN | | | |
| --- | --- | --- | --- | --- | --- | --- | --- | --- |
| | LibriTTS | | CIFAR10 | | LibriTTS | | CIFAR10 | |
| | ASR | TPR/FPR | ASR | TPR/FPR | ASR | TPR/FPR | ASR | TPR/FPR |
| Surrogate model | 89.5 | 32.2/1 | 78.5 | 16.5/1 | 88.3 | 26.2/1 | 76.9 | 19.0/1 |
| Victim model | 89.1 | 32.6/1.1 | 78.3 | 16.8/1.1 | 88.2 | 24.5/0.9 | 76.8 | 19.0/1 |

trend across the different splits. For GradTTS, a similar phenomenon can be observed in Appendix Fig. 6.

**Impact of $\ell_p$-norm** In Fig. 3, we compare the results obtained on $\ell_p$-norm using the $p = 1$ to $7$, with the choice of $t$ being the same as in Section 4.2. The results indicate an increase in performance at $\ell_1$-norm, followed by a decline after the $p = 5$. It reveals that the combined effect of both large and small differences exhibits a synergistic influence when present in an appropriate ratio.

**Determining the value of $\tau$** In Table 4, we present the variation of Attack Success Rate (ASR) and TPR/FPR on the victim model with the $\tau$ determined by the surrogate model. Specifically, we will randomly split the corresponding dataset into two halves four times, resulting in four different train-test splits. We will train four models using these splits. One of the models will be selected as the surrogate model, from which we will obtain the threshold. We will then use this $\tau$ to attack the other three victim models and record the average values. The results indicate that our method achieves promising results when using the $\tau$ selected from the surrogate model.

## 4.4 WHICH TYPE OF MODEL OUTPUT IS MORE ROBUST?

According to Zhang et al. (2023), the TTS pipeline consists of three stages: text to mel-spectrogram, text to audio, and mel-spectrogram to audio. Our experiments tested models for each stage: Grad-TTS for text to mel-spectrogram, DiffWave for mel-spectrogram to audio, and FastDiff for text to audio. There are generally two forms of output: mel-spectrograms and audio. In Table 5, we summarize the model details and best results of our proposed method on three TTS models using the LJSpeech

Table 5: Comparison of different models. AUC is the result on the LJSpeech/TinyImageNet dataset.

| Model | Size | T | Output | Segmentation Length | Best AUC |
| --- | --- | --- | --- | --- | --- |
| DDPM | 35.9M | 1000 | Image | N/A | 92.6 |
| GradTTS | 56.7M | [0, 1] | Mel-spectrogram | 2s | 99.6 |
| DiffWave | 30.3M | 50 | Audio | 0.25s | 52.4 |
| FastDiff | 175.4M | 1000 | Audio | 1.2s | 54.4 |

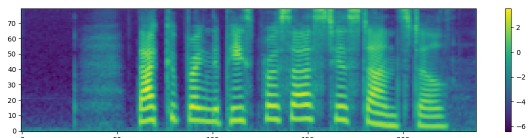

Figure 4: An example of mel-spectrogram.

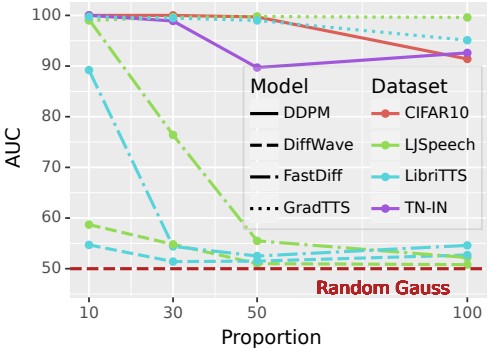

(a) The results of PIA and PIAN on Grad-TTS for different training and evaluation sample numbers.

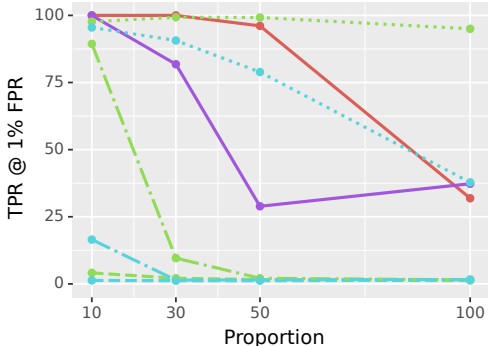

(b) The results of PIA and PIAN on DDPM for different training and evaluation sample numbers.

Figure 5: The performance of our method for different training and evaluation sample numbers. The top row shows the results for AUC, and the bottom row displays the results for TPR @ 1% FPR.

dataset and the DDPM model on the TinyImageNet dataset. We only report the results of our method since it achieves better performance most of the time.

As shown in Table 5, with the same training and hold-out data, GradTTS achieves an AUC close to 100, while DiffWave and FastDiff only achieve the performance slightly above 50, which is close to random guessing. However, DiffWave has a similar size to DDPM and GradTTS, and FastDiff has similar $T$ with DDPM. Additionally, FastDiff has similar segmentation length to GradTTS. Thus, we believe that these hyperparameters are not the decisive parameters for the model's robustness. It is obvious that the output of GradTTS and DDPM is image-like. Fig. 4 provides an example of mel-spectrogram. The deep reasons why these models exhibit robustness can be further explored. We report these results hoping that they may inspire the design of models with MIA robustness.

We also explore the attack performance with various training and evaluation sample numbers. We select 10%, 30%, 50%, and 100% of the samples from the complete dataset. In each split, half of all samples are used for training, and the other half are utilized as a hold-out set. The results are presented in Fig. 5. As we can see, when only 10% of the data is used, relatively high AUC and TPR @ 1% FPR can be achieved. Additionally, we find that the AUC and TPR @ 1% FPR decrease as the proportion of selected samples in the total dataset increases. However, for GradTTS and DDPM, the decrease is relatively gentle, while for DiffWave and FastDiff, the decrease is rapid. In other words, the robustness increases rapidly with the increase of training samples.

## 5    CONCLUSION

In this paper, we propose an efficient membership inference attack method for diffusion models, namely Proximal Initialization Attack (PIA) and its normalized version, PIAN. We demonstrate its effectiveness on a continuous-time diffusion model, GradTTS, and two discrete-time diffusion models, DDPM and Stable Diffusion. Experimental results indicate that our proposed method can achieve similar AUC performance to SecMI and significantly higher TPR @ 1% FPR with the cost of only 2 queries, which is much faster than the 12~62 queries required for SecMI in this paper. Additionally, we analyze the vulnerability of models in TTS, an audio generation task. The results suggest that diffusion models with the image-like output (mel-spectrogram) are more vulnerable than those with the audio output. Therefore, for privacy concerns, we recommend employing models with audio outputs in text-to-speech tasks.

# 6 ACKNOWLEDGMENT

Xiaofeng Zhu was supported in part by the National Key Research & Development Program of China under Grant (No. 2022YFA1004100). Fei Kong and Xiaoshuang Shi were supported by the National Natural Science Foundation of China (No.62276052).

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

APPENDIX

## A  LIMITATION AND BROADER IMPACTS

The purpose of our method is to identify whether a given sample is part of the training set. This capability can be leveraged to safeguard privacy rights by detecting instances of personal information being unlawfully used for training purposes. However, it is important to note that our method could also potentially result in privacy leaking. For instance, this could occur when anonymous data is labeled by determining whether a sample is part of the training set or as a part of data reconstruction attack. It is worth mentioning that our method solely relies on the diffusion model's output as we discussed in the threat model, but it does require the intermediate output. This dependency on the intermediate output may pose a limitation to our method.

## B  DATASETS AND DIFFUSION MODELS

For TTS, we evaluate three commonly used datasets: LJSpeech, VCTK, and a subset of LibriTTS called libritts-lean-100. We test three models: GradTTS [1], FastDiff [2], and DiffWave [3]. For image generation, we evaluate the CIFAR10, CIFAR100 and TinyImageNet datasets using the same DDPM model as Duan et al. (2023), and Laion5, COCO for stable diffusion Rombach et al. (2022). Unless otherwise specified, we randomly select half of the samples as a training set and the other half as the hold-out set.

### B.1  IMPLEMENTATIONS DETAILS

For the audio generation models, we use their codes from the official repositories and apply the default hyperparameters for all models except for the hyperparameters we mentioned. The training iterations were set to 1,000,000, due to the default value for the three audio generative models are all around this. For DDPM, all settings are the same as those in Duan et al. (2023). For various experiments, due to the absence of corresponding trials by the baseline, we employ a grid-search approach to identify the optimal parameters attainable by the method.

Table 6 demonstrate the setting for different attacks. On GradTTS, because SecMI is not designed for continuous-time diffusion, we discretize $[0, 1]$ into 1000 steps and then apply SecMI. We chose $\ell_4$-norm to compute $R_{t,p}$.

To conduct the experiment on stable diffusion, we download the stable-diffusion-v1-5 from [4], without any further fine-tuning or any other modification. We select 2500 sample from 600M laion-aesthetics-v2-5plus as the member set, since stable-diffusion-v1-5 is trained on this dataset as mentioned by HuggingFace. We randomly select 2500 images from the COCO2017-val as the hold-out set, since COCO2017-val is one of the official validation set to examine the performance of stable diffusion. The prompt to generate Laion5 Blip text in BLIP is "A picture of ".

## C  PIA AND PIAN ON GRAD-TTS ACROSS VARIOUS $t$

Fig. 6 demonstrates the results of PIA and PIAN on Grad-TTS for different values of t and different datasets. For GradTTS, consistent with the DDPM, both AUC and TPR @ 1% FPR exhibit a rapid increase at the beginning as $t$ increases followed by a decline around $t = 0.5$.

---

[1] https://github.com/huawei-noah/Speech-Backbones/tree/main/Grad-TTS
[2] https://github.com/Rongjiehuang/FastDiff
[3] https://github.com/lmnt-com/diffwave
[4] https://huggingface.co/runwayml/stable-diffusion-v1-5

| Model | Attack Methol | Attack Time ($t$) |
|-------|---------------|-------------------|
| DDPM | Naive Attack
SecMI
PIA/PIAN | 200
100
200 |
| GradTTS | Naive Attack
SecMI
PIA/PIAN | 0.8
0.6
0.3 |
| Stable Diffusion | Naive Attack
SecMI
PIA/PIAN | 500
100
500 |

Table 6: Summary of attack settings

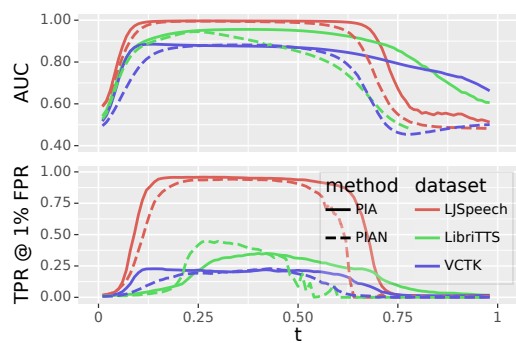

Figure 6: The results of PIA and PIAN on Grad-TTS for different values of $t$ and different datasets.

## D   MORE EXPERIMENTAL RESULTS

### D.1   ROBUSTNESS ON FASTDIFF AND DIFFWAVE

Table 7 shows the AUC of different methods at FastDiff and DiffWave model on three datasets. The performance of all three MIA methods is very poor.

Table 7: Performance of AUC on FastDiff and DiffWave across three datasets.

| Method | FastDiff | | | DiffWave | | |
|--------|----------|------|----------|----------|------|----------|
|        | LJSpeech | VCTK | LibriTTS | LJSpeech | VCTK | LibriTTS |
| NA Matsumoto et al. (2023) | 52.6 | 55.1 | 53.7 | 52.7 | 53.8 | 51.2 |
| SecMI Duan et al. (2023) | 51.6 | 56.3 | 53.7 | 53.2 | 54.3 | 52.4 |
| PIA | 51.6 | 57.1 | 54.1 | 54.4 | 54.2 | 50.8 |
| PIAN | 52.4 | 57.0 | 54.6 | 50.0 | 50.5 | 50.7 |

### D.2   DISTRIBUTION FOR SAMPLES FROM TRAINING SET AND HOLD-OUT SET.

Fig. 7 shows the $R_{t=0.3,p=4}$ distribution for samples from training set and hold-out set at GradTTS on different datasets of PIAN.

### D.3   LOG-SCALED ROC CURVE

As suggested by Carlini et al. (2022), Fig. 8 and Fig. 9 display the log-scaled ROC curves. These curves demonstrate that the proposed method outperforms NA and SecMI at most of times.

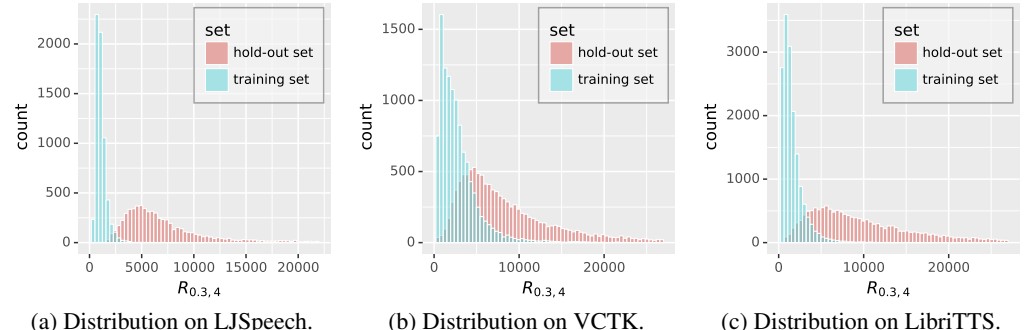

(a) Distribution on LJSpeech.  (b) Distribution on VCTK.  (c) Distribution on LibriTTS.

Figure 7: $R_{t=0.3,p=4}$ distribution for samples from training set and hold-out set at GradTTS on different datasets of PIAN.

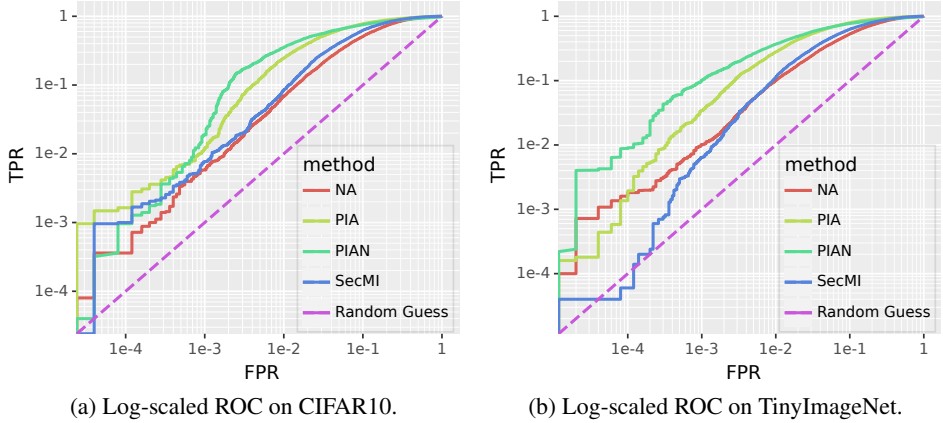

(a) Log-scaled ROC on CIFAR10.  (b) Log-scaled ROC on TinyImageNet.

Figure 8: The log-scaled ROC at DDPM of different methods on CIFAR10 and TinyImageNet.

## D.4 VISUALIZATION OF RECONSTRUCTION

Note Eq. (9) is equal to the distance between $\epsilon_{\theta}(x_0, 0)$ and the predicted one $\epsilon' = \epsilon_{\theta}(x_t, t)$, where $x_t = \sqrt{\bar{a}_t} x_0 + \sqrt{1 - \bar{a}_t} \epsilon_{\theta}(x_0, 0)$. Fig. 10 and Fig. 11 show the reconstructed sample $x_0' = \frac{x_t - \sqrt{1 - \bar{a}_t} \epsilon'}{\sqrt{\bar{a}_t}}$ from $x_t$ using the predicted $\epsilon'$ at DDPM on CIFAR10 of PIAN. The reconstructed samples from $t = 100$ are clear for both the training set and the hold-out set. The reconstructed samples from $t = 400$ are blurry for both sets. However, for $t = 200$, the reconstructed samples are clear for the training set but blurry for the hold-out set.

For GradTTS, we use Eq. (12) to reconstruct samples from $x_t$. This reconstruction is not rigorous, but we just use it to give a visualization. Fig. 12 and Fig. 13 show the reconstructed samples on LJSpeech from PIA. The observed pattern is consistent with DDPM.

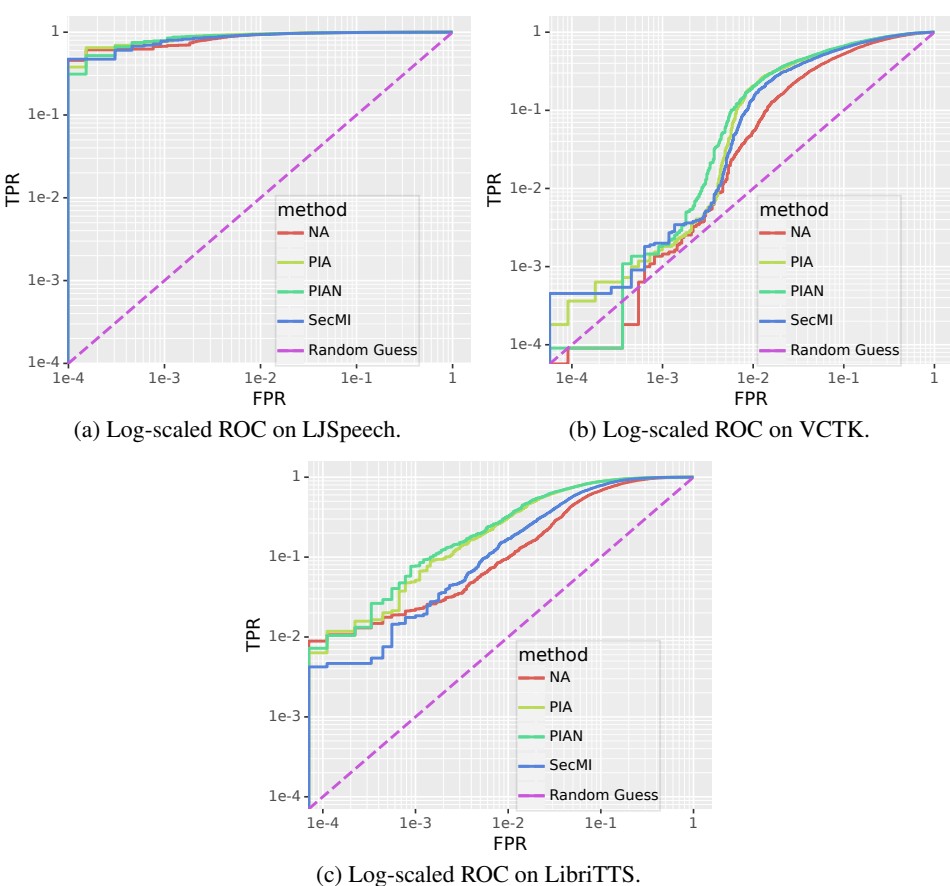

(a) Log-scaled ROC on LJSpeech.

(b) Log-scaled ROC on VCTK.

(c) Log-scaled ROC on LibriTTS.

Figure 9: The log-scaled ROC at GradTTS of different methods on LJSpeech, VCTK and LibriTTS.

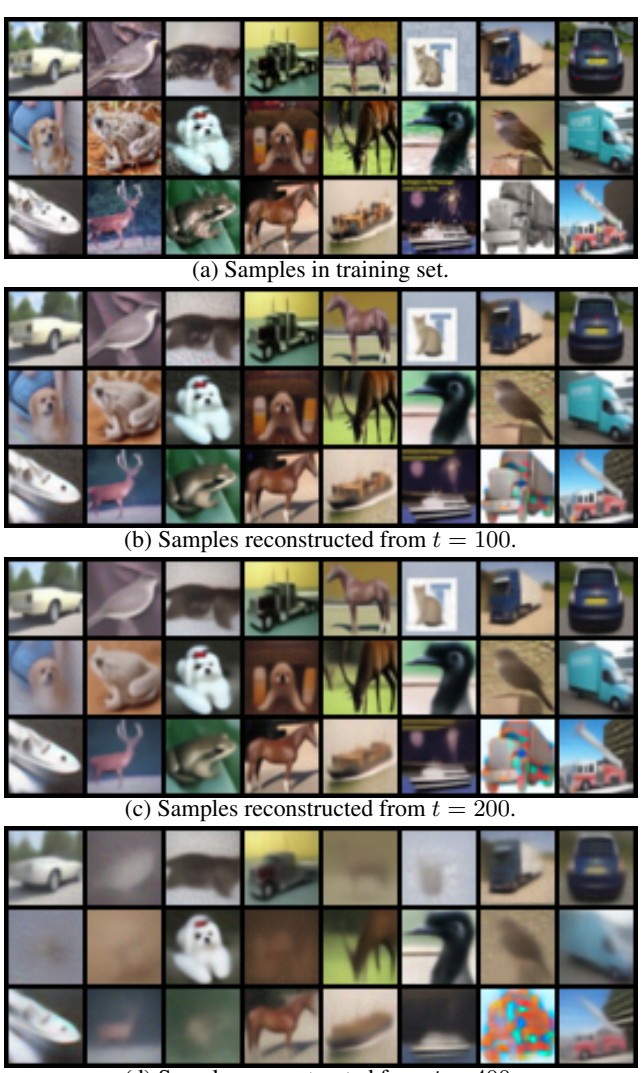

(a) Samples in training set.

(b) Samples reconstructed from $t = 100$.

(c) Samples reconstructed from $t = 200$.

(d) Samples reconstructed from $t = 400$.

Figure 10: Samples in training set and the reconstructed samples at DDPM on CIFAR10 from PIAN.

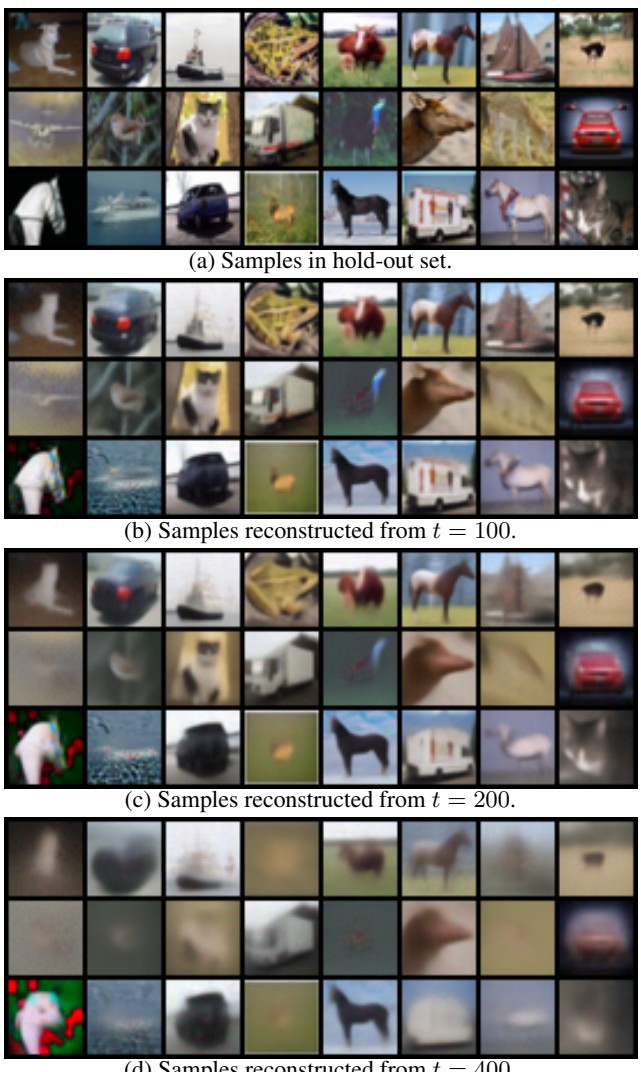

(a) Samples in hold-out set.

(b) Samples reconstructed from $t = 100$.

(c) Samples reconstructed from $t = 200$.

(d) Samples reconstructed from $t = 400$.

Figure 11: Samples in hold-out set and the reconstructed samples at DDPM on CIFAR10 from PIAN.

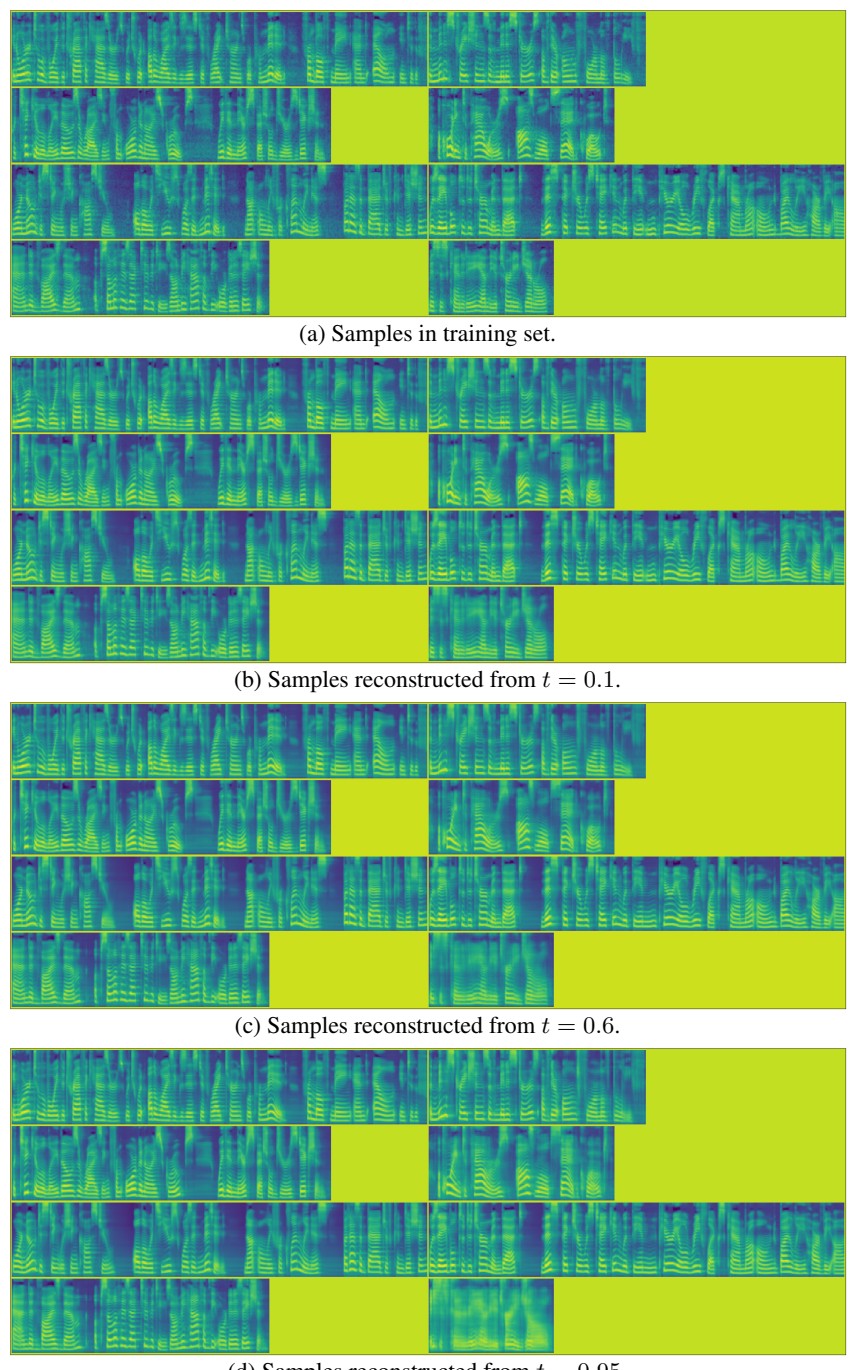

(a) Samples in training set.

(b) Samples reconstructed from $t = 0.1$.

(c) Samples reconstructed from $t = 0.6$.

(d) Samples reconstructed from $t = 0.95$.

Figure 12: Samples in training set and the reconstructed samples at GradTTS on LJSpeech from PIA.

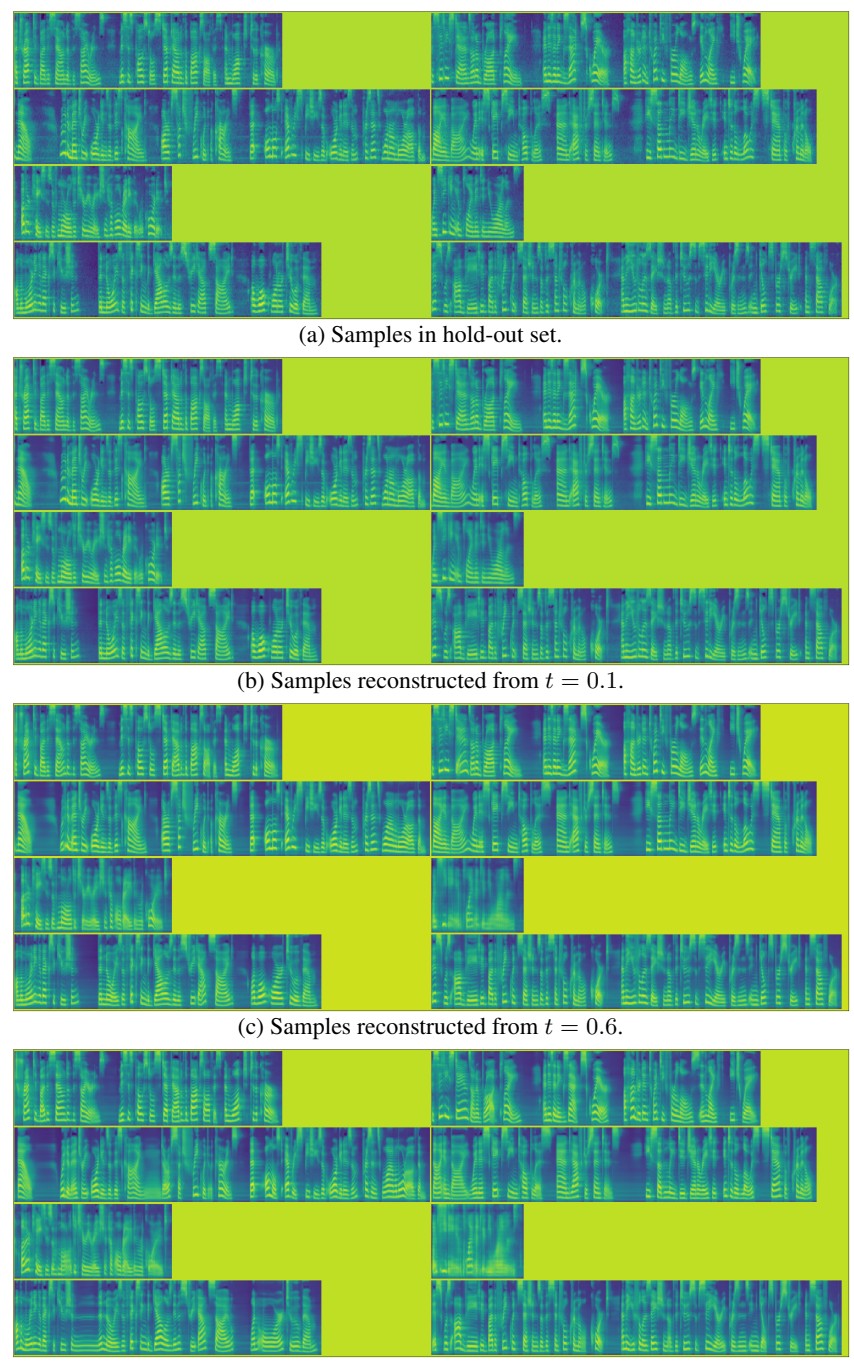

(a) Samples in hold-out set.

(b) Samples reconstructed from $t = 0.1$.

(c) Samples reconstructed from $t = 0.6$.

(d) Samples reconstructed from $t = 0.95$.

Figure 13: Samples in hold-out set and the reconstructed samples at GradTTS on LJSpeech from PIA.

