# OpenReview forum: "An Efficient Membership Inference Attack for the Diffusion Model by Proximal Initialization"
_ICLR.cc/2024/Conference — ICLR 2024 poster_

### Official Review · Reviewer_nuWm · 2023-10-30

**Soundness:** 4 excellent
**Presentation:** 4 excellent
**Contribution:** 3 good
**Rating:** 8
**Confidence:** 4

**Summary:**

This paper proposes a method for conducting a membership inference attack on a diffusion model. Compared to previous methods (SecMI), the proposed method achieves better results while accelerating the process by 5-10 times. The effectiveness of the method is validated on multiple datasets in the paper. Additionally, this paper also analyzed the MIA robustness of diffusion on speech-to-text task, and the authors suggests using models that directly output audio to enhance the MIA robustness of the model.

**Strengths:**

This paper proposed a method achieved a 5-10 times improvement compared to previous SOTA method (SecMI) and conducted extensive experiments to evaluate the proposed method. Strengths are listed below:

1.	This paper conducted extensive experiments, including three image datasets and three audio datasets. In particular, this paper also conducted experiments on stable diffusion, including scenarios where the ground truth text was unknown.
2.	Experiments demonstrated that the proposed method achieved a 5-10 times improvement compared to previous SOTA method (SecMI), while also achieving better results.
3.	This paper analyzed the threat model. When the diffusion model is used for some tasks such as inpainting and classification, it requires the use of intermediate results. In such cases, their method can be employed to attack the model. It’s reasonable.
4.	Experimental results showed that diffusion models directly outputting audio are more robust compared to models outputting mel-spectrograms. This paper recommended using models that directly output audio to enhance robustness.

**Weaknesses:**

The paper is overall clear and sound, but I still have some minor concerns:

1.	When attack audio, this paper also uses these methods proposed from image attacks (NA, SecMI). The methods for attacking audio should be included as well.
2.	In the loss function of the diffusion model, there is a noise term. The proposed method is based on the loss function, but the method to remove the influence of the random seed is not described clearly.
3.	Fig. 2 is too small to read.

**Questions:**

1. It is not clear how hyperparameters are selected for other methods.
2. Why audio data is more robust against the proposed method?
3. It is better to present the procedure of the proposed method in one table.

---

> ### Author Response · Authors · 2023-11-20
>
> **W1: When attack audio, this paper also uses these methods proposed from image attacks (NA, SecMI). The methods for attacking audio should be included as well.**
>
> Thank you for your comment. To our knowledge, no specific attacks target TTS tasks.
>
> We conducted a thorough investigation in the Text-to-Speech domain, particularly on diffusion models, and found no TTS-specific attacks. Consequently, we utilized the state-of-the-art diffusion Membership Inference Attack (MIA) method for images. However, we acknowledge the possibility of overlooking certain methods. If you know of any, we'd be glad to include them in our experiments.
>
> **W2: In the loss function of the diffusion model, there is a noise term. The proposed method is based on the loss function, but the method to remove the influence of the random seed is not described clearly.**
>
> Thanks for your valuable comment. We believe we have removed the influence of random seed.
>
> In fact, although the form of Eq. 9 may appear similar to the training loss, the input for our $\boldsymbol \epsilon$ is the model output at time $t=0$, which results in a deterministic outcome. Furthermore, in the case of continuous time, as seen in Eq. 14, its form is entirely different from the training loss. In addition, the results for DDPM are obtained from the checkpoint and dataset provided in their official repository [r1], while the result of stable diffusion is obtained from the checkpoint provided by HuggingFace [r2]. We also conduct experiments on different dataset splits (Fig. 2). Consequently, we believe we have eliminated the effects of random seed.
>
> **W3: Fig. 2 is too small to read.**
>
> Thanks for your valuable comment. Parts of the subfigures from Figure 2 have been moved to the appendix. They have also been enlarged to improve their readability in the new version.
>
> **Q1: It is not clear how hyperparameters are selected for other methods.**
>
> Thank you for the question. In order to ensure fairness, when conducting experiments, for the same experiment, we choose the default settings in the paper of the method, as well as the checkpoints, data splits, etc. provided by them. For different experiments, especially TTS, we utilize grid-search to find the optimal parameters that the method can achieve. We have put this into Appendix A  in the new version.
>
> **Q2: Why audio data is more robust against the proposed method?**
>
> Thanks for your question. In fact, Models that directly output audio not only exhibit robustness towards our proposed method, but also towards the baseline we have chosen. The output format of the model could be the reason. For example, a sentence breaks down into finite linguistic units or morphemes, and the model might learn their corresponding audios, becoming attack-resistant once all units are learned.
>
> We've analyzed various Text-to-Speech (TTS) pipelines using the same dataset. All models directly creating audio exhibited robustness to MIA, suggesting input type and architecture aren't likely causes. Interestingly, mel-spectrograms resemble one-channel images, hinting again that output type could be a factor. However, this is a hypothesis, and as stated in our paper, further investigation is needed to confirm the reasons behind these observations.
>
> **Q3: It is better to present the procedure of the proposed method in one table.**
>
> Thank you for your suggestion. We have added a table to the paper to make our methods easier to understand in section B of the new version.
>
> [r1] https://github.com/jinhaoduan/SecMI
>
> [r2] https://huggingface.co/runwayml/stable-diffusion-v1-5

---

### Official Review · Reviewer_Vi3v · 2023-10-31

**Soundness:** 4 excellent
**Presentation:** 3 good
**Contribution:** 3 good
**Rating:** 6
**Confidence:** 4

**Summary:**

This paper proposes a novel membership inference attack, Proximal Initialization Attack (PIA), for diffusion models. By using the output at time $t=0$ and calculating the difference between the model outputs at time $t$ and $t-1$, it is possible to determine whether a sample is present in the training dataset. The proposed method is validated for its effectiveness on both image and audio datasets. The results show that compared to the baseline, this method achieves better performance while achieving a 5-10 times speedup.

**Strengths:**

1. This paper aligns with the scope of the conference.
2. The paper is easy to understand, and its motivation is clear.
3. The experiments in the paper are extensive. The proposed method is validated on multiple datasets and multiple models in both image and audio domains, demonstrating its effectiveness. Additionally, the results on the audio dataset indicate that the model with audio output exhibits greater robustness.
4. They demonstrated the scenarios in which the proposed method can be applied (threat model in other word).

**Weaknesses:**

1. Some of the figures in the paper are difficult to read, due to the fact that too many datasets and models are included in the same figure.
2. The pipeline of the TTS tasks is not explained in the paper.
3. Some details of the experimental setup are not clearly described. Specifically, in the experiment on stable diffusion, the checkpoint of BLIP and the specific prompt used are not mentioned in this paper.

**Questions:**

1. How is the query number for the comparison methods determined? Is it possible that the comparison methods used in the article could yield better performance with a lower query number?
2. For the TTS task, this article explores models that generate output in the form of audio and mel-spectrogram. Are there any other types of outputs for the TTS task? It is interesting to investigate whether the other outputs is robust for the TTS task.

---

> ### Author Response · Authors · 2023-11-19
>
> **W1: Some of the figures in the paper are difficult to read, due to the fact that too many datasets and models are included in the same figure.**
>
> Thank you for your suggestion. A subset of the subfigures from Figure 2 has been moved to section B of the appendix. All figures have also been enlarged to improve their readability in the new version.
>
> **W2: The pipeline of the TTS tasks is not explained in the paper.**
>
> Thanks for your valuable comment. We have incorporated the pipeline of the Text-to-Speech (TTS) task into the paper for the readers' understanding in section 4.4 of  the new version.
>
> Specifically, according to reference [r1], the TTS task pipeline can be divided into three parts: text to mel-spectrogram, text to audio, and mel-spectrogram to audio. We have tested models for each pipeline in our experiments. More specifically, Grad-TTS falls under the category of text to mel-spectrogram, DiffWave is categorized under mel-spectrogram to audio, and FastDiff is classified as text to audio.
>
> **W3: Some details of the experimental setup are not clearly described. Specifically, in the experiment on stable diffusion, the checkpoint of BLIP and the specific prompt used are not mentioned in this paper.**
>
> Thanks for your valuable comment. The prompt we used in BLIP is "A picture of ". More specifically, we downloaded the checkpoint from reference [r2], and when inputting each image, we also input the aforementioned prompt.
>
> We have provided more details, including the prompt above, and organized the presentation to make it more readable in the new version.
>
> **Q1: How is the query number for the comparison methods determined? Is it possible that the comparison methods used in the article could yield better performance with a lower query number?**
>
> Thank you for the question. For the NA method, the query number is fixed at 1. However, for SecMI, the query number is directly proportional to the chosen attack parameter $t$. We selected the most effective $t$ through a grid search. Therefore, for SecMI, the performance is worse with the same query number.
>
> **Q2: For the TTS task, this paper explores models that generate output in the form of audio and mel-spectrogram. Are there any other types of outputs for the TTS task? It is interesting to investigate whether the other outputs is robust for the TTS task.**
>
> Thank you for your question.  The types of output are categorized into audio and mel-spectrogram according to [r1]. And to the best of our knowledge, there are no specific attacks targeting TTS task.
>
> We have carried out a meticulous survey in the field of Text-to-Speech focusing on the diffusion model, and as far as we know, there is no specific attacks targeting TTS tasks. Therefore, we have employed the SOTA diffusion Membership Inference Attack (MIA) method used on images. Nevertheless, it is possible that we may have missed some methods. If you are aware of any such methods, we would be delighted to incorporate them into our experiments.
>
>
> [r1] Zhang, C., Zhang, C., Zheng, S., Zhang, M., Qamar, M., Bae, S. H., & Kweon, I. S. (2023). A survey on audio diffusion models: Text to speech synthesis and enhancement in generative ai. arXiv preprint arXiv:2303.13336, 2
>
> [r2] https://huggingface.co/Salesforce/blip-image-captioning-large

---

### Official Review · Reviewer_7qyp · 2023-11-01

**Soundness:** 3 good
**Presentation:** 4 excellent
**Contribution:** 3 good
**Rating:** 8
**Confidence:** 4

**Summary:**

This paper presents a novel approach to membership inference attacks on diffusion models named Proximal Initialization Attack (PIA), focusing on efficiency. The proposed method utilizes the distance between the true trajectory and the estimated trajectory, and the estimated trajectory is obtained by the output of models at $t=0$. With just 2 queries, this approach achieves superior AUC and TPR@1%FPR compared to previous methods.

**Strengths:**

* This paper is well-written and easy to read.
* The experiments are conducted extensively across multiple datasets and models in the domains of images and audio, confirming the effectiveness of the proposed method. A thorough ablation study is also conducted to investigate the patterns of various parameters.
* The results of the experiment are promising, and the experiment is convincing. The proposed method is much faster than the baseline, which is particularly useful for large-scale models.

**Weaknesses:**

* The experimental results are promising. It would be more convincing if you could provide more details, such as the selection of all experimental parameters, including the comparative methods.
* Eq.9 is similar to the original loss function. Is there any connection between the proposed method and the loss function? Please provide further analysis.

**Questions:**

* Why is mel-spectrogram more robust than directly outputting audio?
* Does the model that directly outputs audio have any other costs compared to outputting mel-spectrogram?

---

> ### Author Response · Authors · 2023-11-19
>
> **W1: The experimental results are promising. It would be more convincing if you could provide more details, such as the selection of all experimental parameters, including the comparative methods.**
>
> Thank you for your advice. We have placed most of the experimental settings, including model training details and parameter settings, in Appendix A.1. For comparative methods, the chosen parameters are either the default ones from the original papers or those that produced the best results after grid search. We have provided more details, such as the details for Blip, and organized the presentation to make it more readable in the revision version.
>
> **W2: Eq.9 is similar to the original loss function. Is there any connection between the proposed method and the loss function? Please provide further analysis.**
>
> Thanks for your valuable comment. We believe there is little connection between them.
>
> While the form of Eq. 9 may appear similar to a loss function, there are some distinctions. In the loss function, the input $\boldsymbol\epsilon$ is a Gaussian noise, while in our case, it is the output of the model at time $t=0$. Furthermore, when our method is applied to continuous-time diffusion models, as in Eq. 14, the difference from the training loss becomes evident.
>
> **Q1: Why is mel-spectrogram more robust than directly outputting audio?**
>
> Thank you for your question. The output format of the model could be the reason.
>
> We've analyzed various Text-to-Speech (TTS) pipelines using the same dataset. All models directly creating audio exhibited robustness to MIA, suggesting input type and architecture aren't likely causes. Interestingly, mel-spectrograms resemble one-channel images, hinting again that output type could be a factor. For example, a sentence breaks down into finite linguistic units or morphemes, and the model might learn their corresponding audios, becoming attack-resistant once all units are learned. However, this is a hypothesis, and as stated in our paper, further investigation is needed to confirm the reasons behind these observations.
>
> **Q2: Does the model that directly outputs audio have any other costs compared to outputting mel-spectrogram?**
>
> Thank you for your question. As mentioned in [r1], models that directly output audio are still under development. The cost of employing an end-to-end model lies in the difficulty of achieving better text-to-speech performance compared with GradTTS.
>
>
> [r1] Zhang, C., Zhang, C., Zheng, S., Zhang, M., Qamar, M., Bae, S. H., & Kweon, I. S. (2023). A survey on audio diffusion models: Text to speech synthesis and enhancement in generative AI. arXiv preprint arXiv:2303.13336, 2.

---

### Official Review · Reviewer_Jy2D · 2023-11-03

**Soundness:** 4 excellent
**Presentation:** 3 good
**Contribution:** 4 excellent
**Rating:** 8
**Confidence:** 4

**Summary:**

This paper proposes a novel query-based membership inference attack (MIA), namely Proximal Initialization Attack (PIA), by using groundtruth trajectory obtained by ϵ initialized in t = 0 and predicted point to infer memberships. Experimental results demonstrate the effectiveness of the proposed method on vision and text-to-speech tasks.

**Strengths:**

1. This paper is well-written and clear.
2. The idea is very simple and novel.
3. Very good experimental results.

**Weaknesses:**

1. The motivation of some transformations in the method is not very clear.
2. The third contribution about the experiments is too long and redundant.
3. There exists some grammar errors or typos, like “MIA for generation tasks … has also been extensively researched” in the related work section.

**Questions:**

1. In Eq. (14), the motivation to ignore the high-order infinitesimal term and dt is not clear.
2. Why the proposed method can obtain better performance than the best competitors in Tables 1-3?
3. How to apply the proposed method to discrete-time diffusion models?

---

> ### Author Response · Authors · 2023-11-19
>
> **W1: The motivation of some transformations in the method is not very clear.**
>
>
> R1: Thank you for your insightful comment. Our motivation primarily stems from our fundamental idea.
>
> The concept involves obtaining a sample $\boldsymbol x_t$, using the model to obtain $\boldsymbol x_{t+\Delta}$, and subsequently acquiring $\boldsymbol x_{t}'$. If the model fits this sample better, it is more likely to be within the training set. Simultaneously, as $\Delta t$ increases, the error also increases. Therefore, we chose the smallest value for $\Delta t$. With this choice of $\Delta t$, all our transformations (mainly Eq. 9 and Eq. 13) can be naturally derived.
>
> **W2: The third contribution about the experiments is too long and redundant.**
>
> Thank you for your suggestion. We have revised this part of our contribution in a more concise language.
>
> The modified contribution is below:
> Our evaluations show that PIA matches SecMI's AUC performance and outperforms it in TPR @ 1\% FPR, while being 5-10 times quicker. Moreover, our data imply that in text-to-speech tasks, models producing audio are more resistant to MIA attacks than those generating image-like mel-spectrograms. We therefore suggest using audio-output generation models to minimize privacy risks in audio creation tasks.
>
> **W3: There exists some grammar errors or typos, like “MIA for generation tasks … has also been extensively researched” in the related work section.**
>
> Thanks for your suggestion. We have carefully checked and corrected the grammar errors or typos.
>
> **Q1: In Eq. (14), the motivation to ignore the high-order infinitesimal term and dt is not clear.**
>
> Answer: Thank you for your question. Our motivation is to reduce error during solving ordinary differential equation.
>
> Specifically, our method involves solving the ODE from $\boldsymbol x_t$ to obtain $\boldsymbol x_{t+\Delta t}$ and $\boldsymbol x_{t}'$ respectively. In this process, there is an error that grows with $\Delta t$. Therefore, we choose $\Delta t\rightarrow 0$. In this case, we can ignore the high-order infinitesimals. $\Delta t=dt$. When comparing two samples, we need to choose the same $\Delta t$, so this term can be disregarded.
>
> **Q2: Why the proposed method can obtain better performance than the best competitors in Tables 1-3?**
>
> Thank you for your question. NA uses the training function for the attack, but the noise term in diffusion models may hinder its effectiveness. Both SecMI and our approach replace the noise term with the model's output, thereby obtaining better results. Moreover, our method boosts efficiency by using the output at $t=0$ rather than multi-step iterations, simplifying parameter adjustments and enhancing performance. A distinct distance function from SecMI also contributes to better effectiveness.
>
> **Q3: How to apply the proposed method to discrete-time diffusion models?**
>
> Thanks for your question. As we described in Eq. 9, $R_{t, p}=\left\Vert\epsilon_{\boldsymbol{\theta}}\left(x_{0}, 0\right)-\epsilon_{\boldsymbol{\theta}}\left(\sqrt{\bar a_{t}} x_{0}+\sqrt{1-\bar a_{t}} \epsilon_{\boldsymbol{\theta}}\left(x_{0}, 0\right), t\right)\right\Vert_{p}$, we need to obtain the model's output at time $t=0$. This output is then inserted into $R_{t,p}$. By comparing the value of $R_{t,p}$ with the threshold $\tau$, we can determine whether it is a training sample.

---

### Author Response · Authors · 2023-11-19

Thank you for the positive feedback from all the reviewers and for their constructive suggestions. We have updated our manuscript with the following modifications:


- **Section 1**: We have condensed the third point of contribution.
- **Section 3.4**:  We further elaborate on the derivation of Eq. 14.
- **Section 4.3**: A subset of the subfigures from Figure 2 has been moved to section B. All figures have also been enlarged to improve their readability.
- **Section 4.4**: Some text is added to interpret the pipeline of Text-to-Speech.
- **Section A.1**: We have revised the presentation of the experiment settings to a tabular format and have added further details.

---

### Meta-Review · Area_Chair_2fTs · 2023-12-05

**Metareview:**

The paper proposes a simple approach to extract data from a diffusion model (a.k.a., membership inference attack). The methodology is sound and the approach is simple. The reviewers unanimously agree that the paper is well written.

**Justification For Why Not Higher Score:**

The approach is simple but also lacks depth.

**Justification For Why Not Lower Score:**

The approach is simple. The methodology is sound and well executed. The writing is well.

---

### Decision · Program_Chairs · 2024-01-16

Accept (poster)